

# Effects of low pH and feeding on calcification rates of the cold-water coral *Desmophyllum dianthus*

Ariadna Martínez-Dios[1], Carles Pelejero[1,2], Àngel López-Sanz[1], Robert M. Sherrell[3,4], Stanley Ko[3], Verena Häussermann[5,6], Günter Försterra[5,6] and Eva Calvo[1]

[1] Institut de Ciències del Mar, CSIC, Barcelona, Catalonia, Espanya
[2] Institució Catalana de Recerca i Estudis Avançats, Barcelona, Catalonia, Espanya
[3] Department of Marine and Coastal Sciences, Rutgers, The State University of New Jersey, New Brunswick, NJ, USA
[4] Department of Earth and Planetary Sciences, Rutgers, The State University of New Jersey, Piscataway, NJ, USA
[5] Escuela de Ciencias del Mar/Facultad de Recursos Naturales, Pontificia Universidad Católica de Valparaíso, Valparaíso, Chile
[6] Huinay Scientific Field Station, Huinay, Chile

Corresponding author
Ariadna Martínez-Dios,
amartinez@icm.csic.es

## ABSTRACT

Cold-Water Corals (CWCs), and most marine calcifiers, are especially threatened by ocean acidification (OA) and the decrease in the carbonate saturation state of seawater. The vulnerability of these organisms, however, also involves other global stressors like warming, deoxygenation or changes in sea surface productivity and, hence, food supply via the downward transport of organic matter to the deep ocean. This study examined the response of the CWC *Desmophyllum dianthus* to low pH under different feeding regimes through a long-term incubation experiment. For this experiment, 152 polyps were incubated at pH 8.1, 7.8, 7.5 and 7.2 and two feeding regimes for 14 months. Mean calcification rates over the entire duration of the experiment ranged between −0.3 and 0.3 mg CaCO$_3$ g$^{-1}$d$^{-1}$. Polyps incubated at pH 7.2 were the most affected and 30% mortality was observed in this treatment. In addition, many of the surviving polyps at pH 7.2 showed negative calcification rates indicating that, in the long term, CWCs may have difficulty thriving in such aragonite undersaturated waters. The feeding regime had a significant effect on skeletal growth of corals, with high feeding frequency resulting in more positive and variable calcification rates. This was especially evident in corals reared at pH 7.5 ($\Omega_A = 0.8$) compared to the low frequency feeding treatment. Early life-stages, which are essential for the recruitment and maintenance of coral communities and their associated biodiversity, were revealed to be at highest risk. Overall, this study demonstrates the vulnerability of *D. dianthus* corals to low pH and low food availability. Future projected pH decreases and related changes in zooplankton communities may potentially compromise the viability of CWC populations.

## INTRODUCTION

Over the last two decades, and in parallel with the quantification of the oceanic absorption of anthropogenic $CO_2$ (*Gruber et al., 2019* and references therein), large efforts have been devoted to the study of past, present and future trends in ocean acidification (OA, *Pelejero, Calvo & Hoegh-Guldberg, 2010*; *Gattuso & Hansson, 2011*; *Bopp et al., 2013*). Instrumental time-series, which now cover up to three decades, corroborate that the average surface ocean pH has decreased ~0.1 units since pre-industrial times (*Bates et al., 2014*; *Kapsenberg et al., 2017*). This rate of change is about 100 times faster than during glacial to interglacial transitions (*Pelejero, Calvo & Hoegh-Guldberg, 2010* and references therein). Further declines in seawater pH are projected in the future, accompanied by major changes in marine ecosystems and the organisms that constitute them (*Hurd et al., 2018*). Not surprisingly, both tropical and Cold-Water Corals (CWCs), as marine calcifiers, were the focus of most of the initial research, since the lowering in pH and the associated decrease in carbonate ion ($CO_3^{2-}$) concentration, together with the shoaling of the aragonite saturation state ($\Omega_A$) horizon, could potentially compromise the calcification of their carbonate exoskeletons (*Cohen & Holcomb, 2009*). Cold-water corals comprise many species of ahermatypic corals (Scleractinia, Cnidaria) that generally inhabit high latitude and the deep ocean where specific hydrodynamic and food supply conditions favor coral growth, but that naturally exhibit relatively low values of pH and $\Omega_A$ (*Guinotte et al., 2006*). The future shoaling of the aragonite saturation horizon projected for the global ocean could soon threaten coral provinces in these basins by increasing their exposure to corrosive waters (*Guinotte et al., 2006*; *Touratier & Goyet, 2009*; *Tittensor et al., 2010a*; *Ciais et al., 2013*; *Hassoun et al., 2015*; *Perez et al., 2018*).

Previous studies of tropical corals reported significant decreases in calcification with a lowering of pH (*Chan & Connolly, 2013* and references therein). Similarly, it was expected that OA could impact CWC communities as well (*Guinotte et al., 2006*), and perhaps even more severely, since they are characterized by extraordinary slow-growing and long-lived species that could be vulnerable to rapid and sustained changes in the chemistry of seawater (*Roberts, Wheeler & Freiwald, 2006*). Nevertheless, while there are nearly as many species of CWCs as their shallow tropical counterparts (*Roberts & Cairns, 2014*; *Henry & Roberts, 2017*), little is known about the biology of these enigmatic organisms, and even less about their response to OA. Indeed, culturing studies of CWCs with duration longer than 1 year have scarcely been performed so far, in part because the maintenance of these extremely slow-growing organisms in aquaria is technically challenging but, more importantly, because research on CWCs is a relatively young discipline. The use of remotely operated vehicles has only recently been applied at the depths where these scleractinians often live, generally at a very high associated cost, especially if repeated access or the collection of a large number of individuals is required. This technology provided first glimpses into their biology and ecology and, since then, CWCs are known to contribute to the formation of complex reefs or coral mounds that support highly diverse marine life (*Freiwald & Roberts, 2006*; *Baillon et al., 2012*; *Henry & Roberts, 2017*).

Early studies on effects of OA on CWCs generally targeted key reef-building species such as *Lophelia pertusa* and *Madrepora oculata* and provided diverse and, in some cases, contradictory results. Indeed, many studies showed no significant response in calcification rate under high $p$CO$_2$ (*Maier et al., 2012*, *2013*; *Form & Riebesell, 2012*), but the conclusions were based on incubations shorter than in the present study. More recent literature shows, however, a decrease in net calcification of these species under acidified conditions (*Maier et al., 2016*; *Gómez et al., 2018*). The response of solitary species such as *D. dianthus* has been less explored, but some of the few experiments exhibited signs of physiological stress (*Carreiro-Silva et al., 2014*), especially at certain life stages (*Movilla et al., 2014*). Regarding other stressors in addition to OA, few studies have been carried out with CWCs, but several examined their response to low pH and food availability (*Maier et al., 2016*; *Georgian et al., 2016*; *Büscher, Form & Riebesell, 2017*; *Gómez et al., 2018*). Being sessile suspension feeders, CWCs strongly depend on food resources from the water column, which are often scarce (*Roberts, Wheeler & Freiwald, 2006*), so food availability is very likely a key constraint that modulates the effects of environmental stressors such as OA in these species.

The present study aims to investigate the combined effects of acidification and food availability in the CWC *D. dianthus* by means of a long-term culturing experiment of over 14 months. To the best of our knowledge, this is longer than any other incubation experiment performed so far with CWCs. We carried out the experiment over a wide range of pH (4 levels, 8.1, 7.8, 7.5 and 7.2, thus covering extreme scenarios that may be expected in the future, particularly in the deep Pacific Ocean), and under two feeding regimes. We hypothesized that corals incubated at high frequency feeding would benefit metabolically from the additional energy input, helping them to withstand aragonite undersaturation conditions, especially in the lowest pH treatments. The large sample size of our study (152 polyps) was only possible thanks to the unique conditions in Comau Fjord (Chilean Patagonia), where these corals form dense communities at unusually shallow depths (*Försterra & Häussermann, 2003*; *Cairns, Häussermann & Försterra, 2005*), allowing the collection of a large number of corals of different sizes by means of SCUBA diving. This not only ensured the robustness of our results through replication but also provided the possibility of investigating the response of corals at different life stages.

## MATERIALS AND METHODS

### Studied species

*Desmophyllum dianthus* (Esper, 1794) is a cosmopolitan species of solitary scleractinian coral (Cnidaria, Hexacorallia) living over a wide range of depths (from 7 to more than 4,000 m; *Risk et al., 2002*; *Försterra & Häussermann, 2003*). Most polyps have sizes of 5–10 cm in length and 1.5–3 cm in diameter, although larger specimens (up to 40 cm length) have often been found, especially attached to the underside of rocky overhangs (*Cairns, Häussermann & Försterra, 2005*). Although they form solitary polyps, *D. dianthus* can be found forming dense aggregations of pseudocolonies, especially on vertical rock faces or under overhanging ledges with the calyx facing downwards, presumably to prevent sedimentation, as occurs in Chilean Fjords (*Försterra & Häussermann, 2003*;
*Jantzen et al., 2013b*). *D. dianthus* specimens are considered a generalist heterotrophic species: they feed on a wide range of zooplankton organisms and on a relatively steady supply of particulate organic carbon in the form of marine snow and perhaps also on dissolved organic matter (*Naumann et al., 2011*; *Höfer et al., 2018*).

## Specimen collection and experimental setup

Approximately 200 live polyps of *D. dianthus* were collected manually by means of SCUBA diving in austral summer 2014 at two different sites of Comau Fjord: Punta Huinay and SWALL (Northern Chilean Patagonia; Fig. 1, at ~20 m depth). The collection of animals for scientific purposes at the two sampling sites was approved by the sub-secretariat of fisheries and farming within the Chilean Ministry of Economy, Development & Tourism (ref. 1760). The basaltic walls of the fjord where corals where collected are bathed with seawater of relatively constant temperatures in the 10–12 °C range and salinities of 31–33 psu with a marked and stable low-salinity surface layer, and pH values in the 7.4–8.2 range (Fig. 1; *Försterra et al., 2005*; *Fillinger & Richter, 2013*; *Jantzen et al., 2013a*).

The collected specimens were transported to Barcelona (CITES permit 14CL000006WS) in purpose-designed coolers and, once at the Experimental Aquarium Zone (ZAE) at the Institute of Marine Sciences, they were transferred to large refrigerated seawater containers installed inside a thermostatically controlled room (10 °C) with seawater maintained at this temperature. From the pool of polyps, 152 were carefully cleaned and weighed, and a small lump of epoxy putty was attached to the base of all specimens. In order to simulate their orientation in the natural environment of Comau Fjord, the epoxy bases of the corals were inserted into slots cut into a thick-walled silicone tube that was then positioned in the aquaria so that the corals were "tentacles down" and did not touch any surface or each other. Corals were then distributed in 24 10 L aquaria comprising three replicate aquaria per treatment, with six to seven individual polyps per aquarium, of randomized size ranges, and were left to aclimatise to the experimental conditions for 9 months. During the acclimation process, corals were kept in natural Mediterranean seawater (pH ~8.1) and fed on weekday basis with a mixture of frozen zooplankton (fish food in blister packages, Ocean Nutrition™).

## Carbonate system manipulation

After the acclimation period, we implemented a pH-manipulative experimental system following the experimental design described in *Movilla et al. (2012)*. The natural Mediterranean seawater was obtained from an underwater intake at 10 m depth and 300 m offshore, and then 50 and 10 μm filtered. A continuous flow of this water was supplied to four 150 L tanks using a diaphragm pump (Iwaki IX-C150TCR-TB1-E, 60 L/h) where pH was adjusted to values of ~8.1, ~7.8, ~7.5 and ~7.2 (on the total scale, pHT; Fig. 2) providing saturation state of aragonite ($\Omega A$) of ~2.7, ~1.3, ~0.8 and ~0.4, respectively. Although these chosen pH and $\Omega_A$ levels may seem rather extreme, projections using the most pessimistic emission scenario RCP8.5 (Representative Concentration Pathway 8.5, where 8.5 is the radiative forcing, in W/m², expected by the year 2100; *Van Vuuren et al.,*
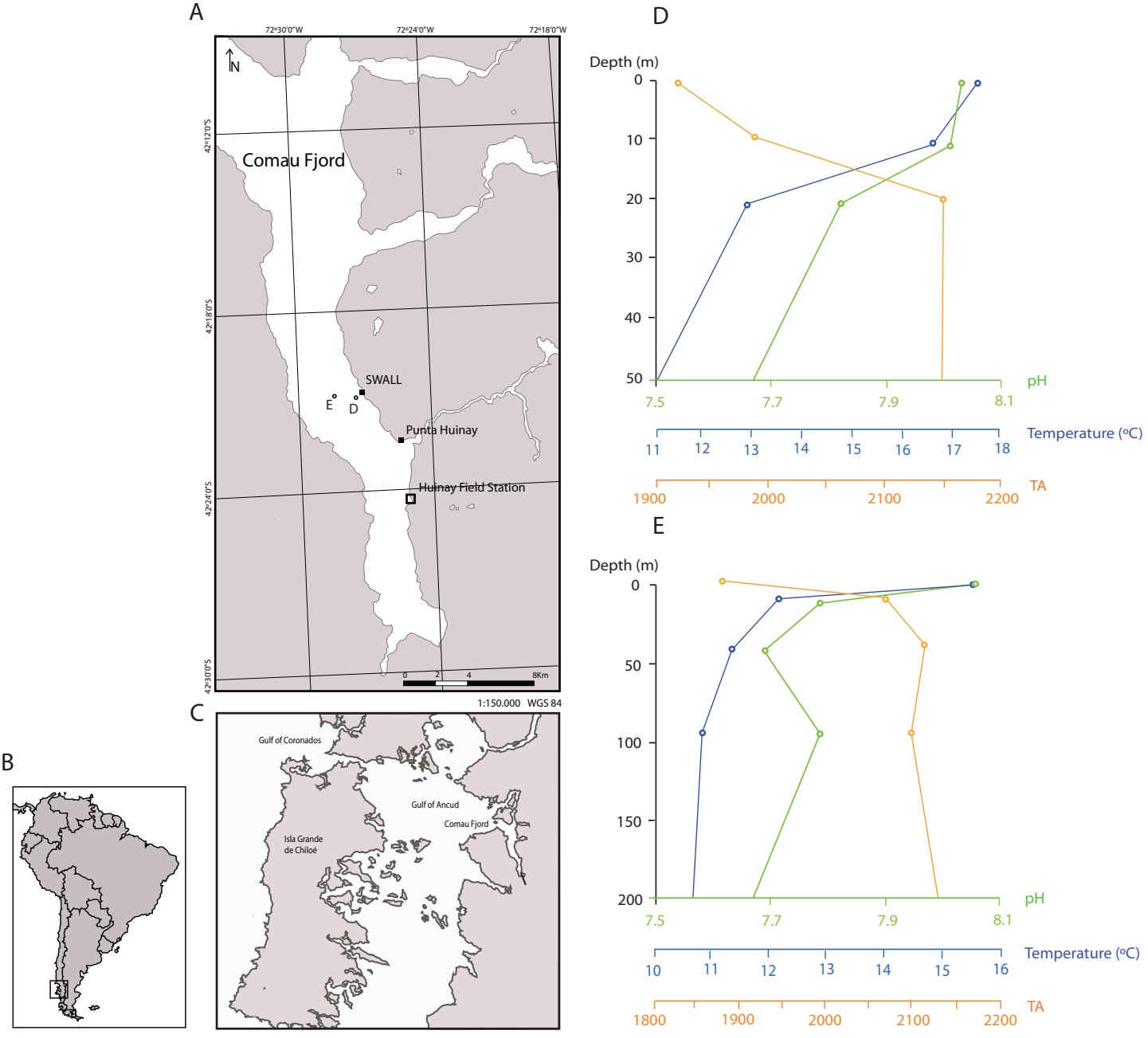

**Figure 1 Sampling locations in the Comau Fjord, Chilean Patagonia and depth profiles of water column properties.** (A) All polyps used in the experiment were sampled at the sites SWALL and Punta Huinay. (B) and (C) Geographical context. (D) and (E) show depth profiles of temperature, pH (total scale) and alkalinity (μmol/kg) close to collection sites (*Jantzen et al., 2013b*; see (D) and (E) in the map) indicating that conditions at the collection depth (~20 m) agree with conditions set in the experiment regarding temperature and pH (T° = 11 °C; pH = 8, total scale). There was, however, a significant difference in alkalinity, fundamentally due to the difference in salinity between the Comau Fjord and Mediterranean water. Nevertheless, all polyps adapted rapidly to the new environmental conditions, actively opening their tentacles as in the field.

*2011*), reach values of $\Omega_A$ below 0.5, particularly in the deep Pacific Ocean (*Ciais et al., 2013*). In the four large tanks, we either bubbled $CO_2$ (99.9% purity) or $CO_2$-free air (using a home-made filter filled with soda lime, Sigma Aldrich, St. Louis, Missouri, USA) to
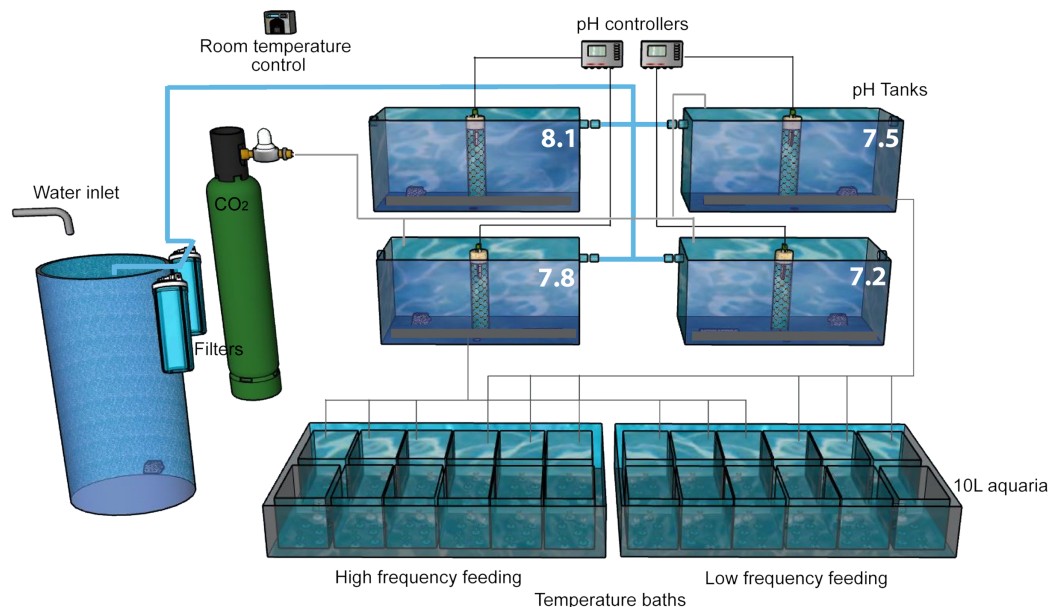

**Figure 2 Experimental setup.** The experiment was performed in a thermostatically controlled room at 10 °C. The aquaria were maintained with a continuous flow of filtered (50 and 10 µm) Mediterranean seawater. The pH was adjusted at the four head tanks with glass electrodes connected to pH controllers which automatically opened and closed solenoid valves of $CO_2$ or $CO_2$-free air as needed. Water in all 10 L aquaria overflowed to the large temperature baths, each of which had a water outlet (not shown).

reduce or increase pH, respectively. Water from each large tank was transferred continuously to three replicates 10 L experimental aquaria for each feeding level, where the corals were maintained. Aquaria were placed in a thermo-regulated bath in order to prevent temperature oscillations and differences between aquaria. Water movement in the aquaria was provided by recirculation pumps and seawater renewal rate in these aquaria was five times per day. Additionally, all the aquaria were covered with a loose-fitting methacrylate cap to reduce evaporation and minimize surface-air gas exchange, while allowing seawater to overflow the lip of the aquarium. During the first week of the acidification experiment, the pH was adjusted gradually (0.03 units per day) to the final selected values and maintained there for 433 days.

Seawater pH was monitored continuously by glass electrodes (LL Ecotrode plus—Metrohm) connected to a pH controller (Consort R305 and R362; Topac Inc., Cohasset, MA, USA), which automatically opened and closed the solenoid valves of $CO_2$ or $CO_2$-free air when needed. To avoid drift in the pH measurements, glass electrodes were calibrated on a daily basis with a TRIS buffer, following standard procedures (SOP6a of *Dickson, Sabine & Christian (2007)*). Temperature and salinity in the aquaria were measured every 2–3 days using an YSI-30 M/10FT probe. In addition, analyses of alkalinity (TA) by potentiometric titration (*Perez & Fraga, 1987*; *Perez et al., 2000*) and seawater pH by spectrophotometry (*Clayton & Byrne, 1993*) were carried out monthly. Measurements (TA, pH, salinity and temperature) were used to calculate other variables of the $CO_2$ system including dissolved inorganic carbon (DIC), carbonate ion concentration ($CO_3^{2-}$),

**Table 1 Chemical and physical properties of seawater in the aquaria for each treatment (mean ± SD) during the experiment.**

**(A) Measured temperature, salinity, pH and alkalinity (TA).**

| Measured variables | | T (°C) | Salinity | pH (total scale) | TA (µEq/kg) |
|---|---|---|---|---|---|
| HF | 8.1 | 11.0 ± 0.1 | 37.9 ± 0.15 | 8.12 ± 0.01 | 2,537 ± 4 |
| | 7.8 | 11.1 ± 0.2 | 37.9 ± 0.15 | 7.73 ± 0.05 | 2,541 ± 4 |
| | 7.5 | 11.0 ± 0.1 | 37.9 ± 0.15 | 7.50 ± 0.02 | 2,539 ± 4 |
| | 7.2 | 11.0 ± 0.1 | 37.9 ± 0.15 | 7.25 ± 0.02 | 2,540 ± 4 |
| LF | 8.1 | 10.9 ± 0.1 | 37.9 ± 0.15 | 8.10 ± 0.03 | 2,537 ± 4 |
| | 7.8 | 10.9 ± 0.1 | 37.9 ± 0.15 | 7.73 ± 0.05 | 2,541 ± 4 |
| | 7.5 | 10.9 ± 0.1 | 37.9 ± 0.15 | 7.52 ± 0.01 | 2,539 ± 4 |
| | 7.2 | 10.9 ± 0.1 | 37.9 ± 0.15 | 7.24 ± 0.02 | 2,541 ± 4 |

**(B) Carbonate system in the aquaria**

| Calculated variables | | $p\mathrm{CO_2}$ (µatm) | $\chi\mathrm{CO_2}$ (ppm) | DIC (µmol/kg) | $[\mathrm{HCO_3^-}]$ (µmol/kg) | $[\mathrm{CO_3^{2-}}]$ (µmol/kg) | $[\mathrm{CO_2}]_{aq}$ (µmol/kg) | $\Omega_A$ |
|---|---|---|---|---|---|---|---|---|
| HF | 8.1 | 360 ± 10 | 360 ± 10 | 2,274 ± 8 | 2,073 ± 10 | 187 ± 3 | 14.8 ± 0.3 | 2.79 ± 0.04 |
| | 7.8 | 1,030 ± 140 | 1,040 ± 140 | 2,453 ± 21 | 2,322 ± 24 | 88 ± 9 | 42.6 ± 5.2 | 1.31 ± 0.13 |
| | 7.5 | 1,730 ± 100 | 1,750 ± 100 | 2,533 ± 10 | 2,408 ± 9 | 53 ± 3 | 72.0 ± 3.8 | 0.79 ± 0.04 |
| | 7.2 | 3,180 ± 190 | 3,220 ± 190 | 2,628 ± 9 | 2,465 ± 4 | 30 ± 2 | 132.4 ± 7.4 | 0.45 ± 0.02 |
| LF | 8.1 | 384 ± 34 | 389 ± 35 | 2,287 ± 16 | 2,092 ± 23 | 179 ± 8 | 16.0 ± 1.2 | 2.67 ± 0.13 |
| | 7.8 | 1,010 ± 130 | 1,030 ± 130 | 2,453 ± 20 | 2,323 ± 22 | 88 ± 8 | 42.4 ± 5.0 | 1.31 ± 0.12 |
| | 7.5 | 1,616 ± 37 | 1,637 ± 37 | 2,525 ± 6 | 2,403 ± 5 | 55 ± 1 | 67.5 ± 1.3 | 0.82 ± 0.02 |
| | 7.2 | 3,190 ± 190 | 3,230 ± 190 | 2,630 ± 8 | 2,467 ± 3 | 30 ± 2 | 133.1 ± 6.8 | 0.44 ± 0.02 |

Note:

$p\mathrm{CO_2}$, partial pressure of $\mathrm{CO_2}$; $\chi\mathrm{CO_2}$, molar fraction of $\mathrm{CO_2}$; $[\mathrm{HCO_3^-}]$, bicarbonate ion concentration; $[\mathrm{CO_3^{2-}}]$, carbonate ion concentration; $[\mathrm{CO_2}]_{aq}$, concentration of dissolved $\mathrm{CO_2}$; $\Omega_A$, aragonite saturation state were calculated using the CO2SYS Excel spreadsheet software (*Pierrot, Lewis & Wallace, 2006*) and the measured variables.

bicarbonate ion concentration ($\mathrm{HCO_3^-}$), partial pressure of $\mathrm{CO_2}$ ($p\mathrm{CO_2}$), aragonite saturation state ($\Omega_A$) and the mole fraction of $\mathrm{CO_2}$ in dry air ($\chi\mathrm{CO_2}$), using the CO2SYS excel spreadsheet software (*Pierrot, Lewis & Wallace, 2006*) with dissociation constants for carbonate species determined by *Mehrbach et al. (1973)* and refit by *Dickson & Millero (1987)*. Chemical and physical conditions of both treatments during the experiment are shown in Table 1.

This experimental setup was developed in duplicate to allow for two feeding treatments at the same time (Fig. 2). In order to simulate as well as possible the natural field conditions and to compare with published literature, corals in this experiment were fed taking into account both the food type and the optimal prey size observed in natural habitats of *D. dianthus* (*Höfer et al., 2018*; *Sánchez, González & Iriarte, 2011*), in trophic experiments with different CWCs (*Tsounis et al., 2010*; *Purser et al., 2010*; *Naumann et al., 2011*; *Larsson, Lundälv & Van Oevelen, 2013*) and our own expertise with previous experiments with this species (*Movilla et al., 2014*). Corals were fed with a mixture of frozen zooplankton (fish food in blister packages, Ocean Nutrition™) that included copepods of the genus *Cyclops* (one cube), and small crustaceans such as *Artemia salina* (Crustacea, Sarsostraca) alternated with one cube of *Mysis relicta* (Crustacea, Eumalacostraca), all of them resuspended together in 40 mL of seawater. High frequency feeding (HF) corals were

fed 5 days per week (Monday to Friday) while low frequency feeding (LF) corals were fed only twice per week (Monday and Thursday) using the same diet. The same volume (3 mL) of resuspended food was aliquotted to each aquarium using a graduated pipette. The average number of food items supplied to each aquarium at each feeding was assessed with repeated counts using an Olympus Stereomicroscope SZ60 under ×1 magnification with a Bogorov chamber (*Gannon, 1971*). The average concentration of food particles in each 10 L aquarium after each food delivery was approximately 91 cyclops, three artemia and two mysis per liter (Table S1).

## Skeletal growth rates

Skeletal growth of all coral polyps was assessed by means of the buoyant weight (BW) technique (*Jokiel, Maragos & Franzisket, 1978*; *Davies, 1989*), using a 0.1 mg resolution balance (Mettler Toledo AB204 SFACT). An initial measurement of the BW was performed at the time when the organisms were distributed in the aquaria (T = −1), a second one when the experiment started (T = 0), and subsequent weighing was conducted every 3 to 7 months throughout the experiment: T1 (119 days from T0), T2 (221 days from T0) and T3 (433 days from T0). During these measurements, temperature and salinity of seawater were monitored continuously using a YSI-30M probe. The net BW of the corals (calculated as the total coral weight minus the weight of the coral holder) was transformed to dry weight using the density of aragonite skeleton for *D. dianthus* species (2.78 g/cm$^3$; *Movilla et al., 2014*). Calcification rates were normalized to the skeletal weight at the beginning of the experiment (T0) and subsequently, to the beginning of each sampling period (T1–T3). Results are expressed as the increase in mg CaCO$_3$ g$^{-1}$ d$^{-1}$, which can be translated easily into % mass increase d$^{-1}$ for comparison with other published studies.

## Statistical analyses

The underlying questions of our study, whether there exists a relationship between skeletal accretion and seawater pH, and whether this relationship is modulated by the feeding regime, were evaluated statistically. For this analysis, we used calcification rate data for the 152 corals at the three discrete BW measurements. Moreover, since *D. dianthus* species are solitary corals that do not form colonies, all corals were treated as different individuals for statistical inference. Normality and homogeneity of variance were assessed graphically with QQ plots of residuals and Cook's distance plots to confirm analytical assumptions. Subsequently, a generalized least of squares (GLS) model was applied to untransformed data with the function *gls* (package nlme, *Pinheiro et al., 2018*) to evaluate the statistical significance of the experimental conditions (pH, feeding frequency, incubation time and coral's initial size) on net calcification rate. Treatments were used as fixed effects while time and aquaria within each treatment were set as random effects in the statistical model with the formula: ~1|time*aquaria, which adjusts for different variance per stratum. Differences among groups were examined post hoc with the *lsmeans* function (package lsmeans, (*Lenth, 2016*)). All the analyses were performed using the RStudio 1.1.383 software

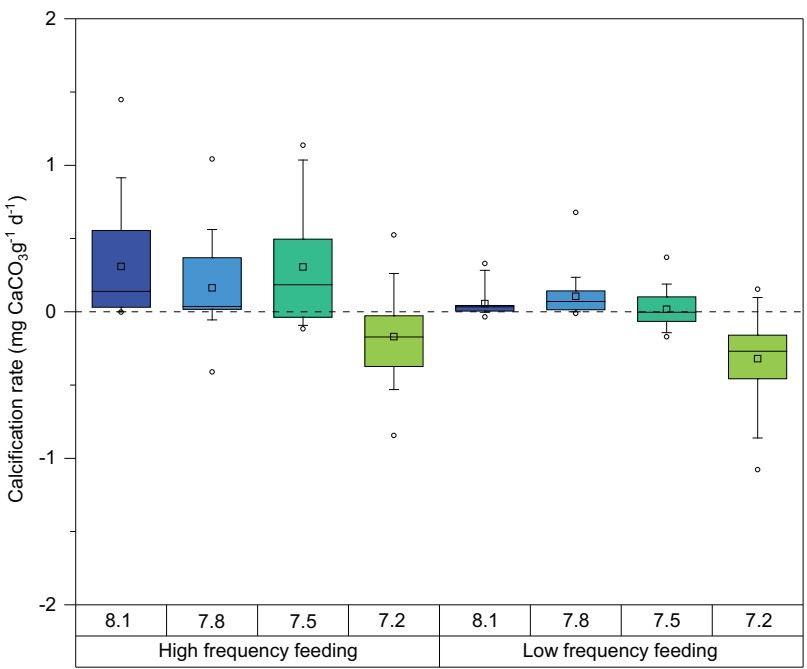

**Figure 3 Box-and-whisker plots of skeletal growth rates of *D. dianthus* specimens during the whole experiment (433 days of incubation) at the four pH levels and two frequency feeding treatments.** Boxes delimit the first and third quartiles, with a horizontal line at the median, and whiskers represent the 10th and 90th percentiles. Outliers are represented by empty circles. The total number of polyps was 152.

(2009–2017 RStudio, Inc, Boston, MA, USA). Data are reported as untransformed means and standard errors.

## RESULTS

### Carbonate system parameters

The seawater $CO_2$ system variables of all treatments were calculated from periodic paired spectrophotometric $pH_T$ and TA laboratory measurements (Table 1). For all treatments, temperature, salinity and alkalinity remained virtually constant for the duration of the experiment ($10.0 \pm 0.1$ °C, $37.9 \pm 0.2$ and $2{,}539 \pm 4$ μmol/kg, respectively). The pH in all treatments was kept reasonably constant over the entire experiment, with values of $8.11 \pm 0.03$, $7.73 \pm 0.05$, $7.50 \pm 0.02$ and $7.24 \pm 0.02$ (mean ± SE). These levels led to $\Omega_A$ values of $2.74 \pm 0.13$, $1.31 \pm 0.13$, $0.80 \pm 0.04$ and $0.45 \pm 0.02$, respectively.

### Skeletal growth rates

Mean calcification rates during the entire experiment (T0–T3) ranged between −0.3 and 0.3 mg $CaCO_3$ $g^{-1}d^{-1}$ (Fig. 3; Table 2). The explanatory variables to major observed differences in growth, evident in Fig. 3, were pH (ANOVA $F = 105.22$, $p$-value ≤ 0.0001, DF = 436) and feeding regime (ANOVA $F = 4.088$, $p$-value = 0.0438, DF = 436) (Table 3).

In general terms, as observed in Fig. 3, HF corals showed higher calcification rates than LF corals as well as larger variability. Significant differences were also observed between feeding regimes, in all pH treatments. Calcification rates of corals at pH 8.1 were very

**Table 2 Descriptive statistics of the skeletal growth by treatment for the whole experiment (433 days).** Mean calcification rate expressed as mg CaCO$_3$ g$^{-1}$d$^{-1}$. $n = 152$.

| | pH | Calcification rate | | | | | | | | | | Dissolution | |
|---|---|---|---|---|---|---|---|---|---|---|---|---|---|
| | | Maximum | Median | Mean | % d$^{-1}$ | SD | SE | Minimum | Variance | Skewness | Kurtosis | N | % Affected |
| HF | 8.1 | 1.448 | 0.139 | 0.309 | 0.031 | 0.39 | 0.0089 | −0.003 | 0.15 | 1.75 | 2.92 | 0 | 0 |
| | 7.8 | 1.043 | 0.037 | 0.163 | 0.016 | 0.31 | 0.0072 | −0.410 | 0.10 | 1.23 | 2.61 | 2 | 11 |
| | 7.5 | 1.136 | 0.185 | 0.305 | 0.033 | 0.40 | 0.0101 | −0.116 | 0.16 | 0.95 | −0.22 | 5 | 26 |
| | 7.2 | 0.524 | −0.172 | −0.170 | −0.015 | 0.31 | 0.0067 | −0.845 | 0.10 | 0.19 | 0.71 | 15 | 79 |
| LF | 8.1 | 0.329 | 0.034 | 0.055 | 0.006 | 0.09 | 0.0021 | −0.035 | 0.01 | 2.34 | 4.98 | 1 | 5 |
| | 7.8 | 0.678 | 0.070 | 0.105 | 0.011 | 0.15 | 0.0035 | −0.011 | 0.02 | 3.15 | 11.56 | 1 | 5 |
| | 7.5 | 0.372 | −0.003 | 0.017 | 0.002 | 0.13 | 0.0031 | −0.171 | 0.02 | 1.05 | 1.42 | 8 | 42 |
| | 7.2 | 0.154 | −0.270 | −0.320 | −0.032 | 0.31 | 0.0070 | −1.078 | 0.09 | −0.88 | 1.10 | 13 | 68 |
| Total | | | | | | | | | | | | 45 | 30 |

**Table 3 The effect of experimental conditions on the calcification rate of *Desmophyllum dianthus* corals was tested taking as a reference the calcification rate at each sampling period (discrete successive measures T0–T3).** Note that the number of values that are free to vary, known as degrees of freedom (DF, $n-1$) in this model, refers then to 152-1 corals per three discrete measurements.

| Factors | DF | F-value | p-value |
|---|---|---|---|
| Intercept | 1 | 42.757 | <0.0001 |
| pH | 3 | 88.325 | <0.0001 |
| Feed | 1 | 26.548 | <0.0001 |
| Weight class | 3 | 21.876 | <0.0001 |
| Time | 2 | 2.800 | 0.0619 |
| Residual standard error | 0.6751729 | | |
| Degrees of freedom: | 446 total | 436 residual | |

similar to those at 7.8 and started to exhibit some detrimental effects at pH 7.5. Interestingly, the response of calcification was more acute in LF corals (Fig. 3). At pH 7.2, decreases in calcification were evident from the beginning of the experiment (Table 4). Indeed, during weight measurements, visual inspection of corals in this treatment revealed increased mucus production and retraction of the basal coenosarc tissue, resulting in the direct exposure of the skeleton to seawater.

These observations were statistically significant, and post hoc analysis on the marginal means indeed revealed significant differences at the 95% confidence level on the calcification rate among pH treatments (Table 5), especially between the pH 7.2 and 8.1 corals (TUKEY HSD $t$-ratio = −15.736, $p$-value ≤ 0.0001). In fact, calcification was on average 17% and 30% lower in the 7.2 treatment in the HF and LF treatments, respectively (Fig. 3; Table 2). As mentioned above, boxplots in Fig. 3 denote that corals incubated at pH 7.8 barely differed from those exposed at pH 8.1 (TUKEY HSD $t$-ratio = 1.116, $p$-value = 0.68) (Table 5).

The effect of incubation time on growth was examined and found to be statistically significant (ANOVA $F$ = 2.800, $p$-value = 0.0619). As observed in Table 4, all treatments

**Table 4 Calcification rates between periods.** Mean calcification rate and its standard deviation expressed as mg $CaCO_3$ $g^{-1}d^{-1}$, percentage mass increase and standard deviation by periods of measurement (discrete weight measurements T0–T3). $n = 152$.

| | pH | T0–1 | | | T1–2 | | | T2–3 | | |
|---|---|---|---|---|---|---|---|---|---|---|
| | | Mean | % d$^{-1}$ | SD | Mean | % d$^{-1}$ | SD | Mean | % d$^{-1}$ | SD |
| HF | 8.1 | 0.348 | 0.035 | 0.53 | 0.340 | 0.031 | 0.41 | 0.272 | 0.023 | 0.36 |
| | 7.8 | 0.237 | 0.024 | 0.63 | 0.136 | 0.012 | 0.25 | 0.142 | 0.013 | 0.33 |
| | 7.5 | 0.360 | 0.036 | 0.85 | 0.250 | 0.024 | 0.45 | 0.310 | 0.027 | 0.42 |
| | 7.2 | −0.279 | −0.028 | 0.30 | −0.266 | −0.028 | 0.30 | −0.063 | −0.008 | 0.44 |
| LF | 8.1 | 0.067 | 0.007 | 0.13 | 0.069 | 0.007 | 0.11 | 0.042 | 0.004 | 0.10 |
| | 7.8 | 0.124 | 0.012 | 0.35 | 0.048 | 0.005 | 0.06 | 0.128 | 0.012 | 0.14 |
| | 7.5 | 0.001 | 0.000 | 0.28 | 0.000 | 0.000 | 0.11 | 0.034 | 0.003 | 0.17 |
| | 7.2 | −0.400 | −0.040 | 0.52 | −0.391 | −0.041 | 0.27 | −0.241 | −0.029 | 0.36 |

**Table 5 Results of the post hoc pair-wise comparisons on the estimated marginal means.**

| Contrast | Estimate | SE | df | t. ratio | p-value |
|---|---|---|---|---|---|
| High frequency feeding | | | | | |
| 7.2–7.5 | −0.2371 | 0.0213 | 61.3 | −11.140 | <0.0001 |
| 7.2–7.8 | −0.3331 | 0.0209 | 65.3 | −15.914 | <0.0001 |
| 7.2–8.1 | −0.3206 | 0.0204 | 56.8 | −15.736 | <0.0001 |
| 7.5–7.8 | −0.096 | 0.0127 | 42.9 | −7.563 | <0.0001 |
| 7.5–8.1 | −0.0835 | 0.0108 | 26.2 | −7.718 | <0.0001 |
| 7.8–8.1 | 0.0124 | 0.0112 | 72.8 | 1.116 | 0.6811 |
| Low frequency feeding | | | | | |
| 7.2–7.5 | −0.2371 | 0.0213 | 61.3 | −11.140 | <0.0001 |
| 7.2–7.8 | −0.3331 | 0.0209 | 65.3 | −15.914 | <0.0001 |
| 7.2–8.1 | −0.3206 | 0.0204 | 56.8 | −15.736 | <0.0001 |
| 7.5–7.8 | −0.096 | 0.0127 | 42.9 | −7.563 | <0.0001 |
| 7.5–8.1 | −0.0835 | 0.0108 | 26.2 | −7.718 | <0.0001 |
| 7.8–8.1 | 0.0124 | 0.0112 | 72.8 | 1.116 | 0.6811 |

showed a slight reduction in calcification with time and, after 433 days of exposure, the average calcification was 11% lower compared to the beginning of the experiment.

When comparing the cumulative skeletal growth rate (in mg CaCO3 $g^{-1}$ $d^{-1}$, normalized to the total polyp mass) for the whole experiment against the initial weight of each polyp at the beginning of the experiment (T0), there is a tendency for smaller corals to exhibit greater calcification rates than those with a higher initial weight (Fig. 4). Calcification rates of smaller corals were affected to a greater relative degree at the lowest seawater pH. The survivorship in each treatment was 100% with the exception of the pH 7.2 treatment aquaria, where 30% of corals died during the incubation regardless of the feeding. Moreover, as shown in Fig. 5, negative calcification rates were detected for a substantial number of corals and visual inspection of the corals revealed the deterioration of the skeletons. This occurred in 31% of the total number of individuals, with those

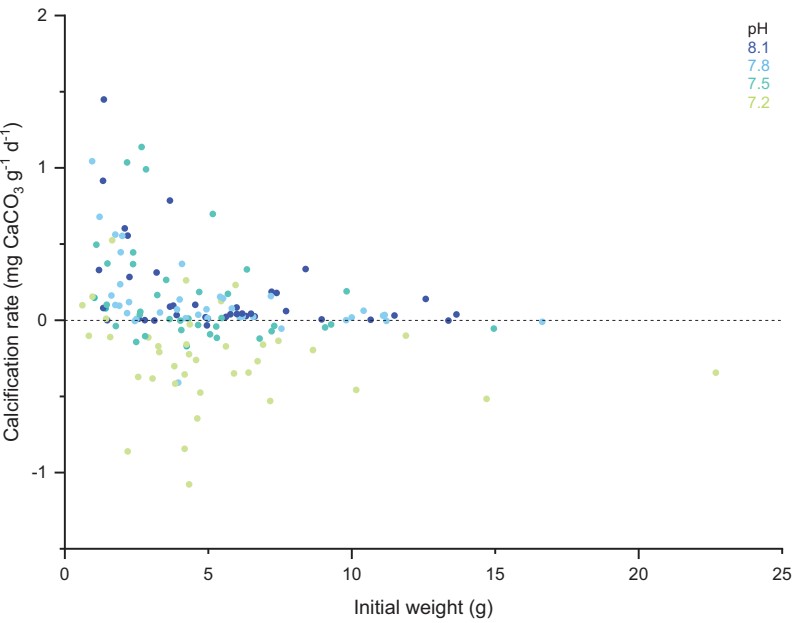

**Figure 4 Scatter plot of the initial weight of the polyps and their calcification rate computed for the total duration of the experiment (433 days).** Note the color code corresponding to the different pH treatments.

exposed to pH 7.2 the worst affected; ~73% of the corals in this treatment showed clear signs of skeletal degradation such as a more fragile, porous skeleton, some with small holes.

## DISCUSSION

In this study, we report data from a 14 month OA culturing experiment with the CWC *D. dianthus*. In addition to pH manipulation, we have performed the experiment at two different frequency feeding regimes, to assess whether increased food supply could mitigate or partially offset the adverse effects of OA. With this experimental setup, we extract conclusions regarding some of the key conditions (seawater pH and feeding) for optimum growth in this CWC species at different life stages.

### pH and aragonite saturation state ranges for optimum growth

Our results indicate a strong detrimental effect of sustained exposure to low pH (7.5 and below) and aragonite undersaturation ($\Omega_A < 1$) on calcification rates of *D. dianthus*. The observed mortality and negative calcification rates (i.e., more dissolution than calcification), especially on polyps grown at pH 7.2, suggests that, in the long term, CWCs will not be able to continue growing and maintaining their skeletons under these conditions. Our experiment suggests that, under $\Omega_A$ values of around 0.8, *D. dianthus* may not be able to calcify properly, depending on food intake rate, which in our experiment and others has not been quantified in absolute terms. Indeed, a threshold between positive and negative balance of calcification and dissolution was also found in the CWC *M. oculata* at $\Omega_A$ of 0.92 (*Maier et al., 2016*).

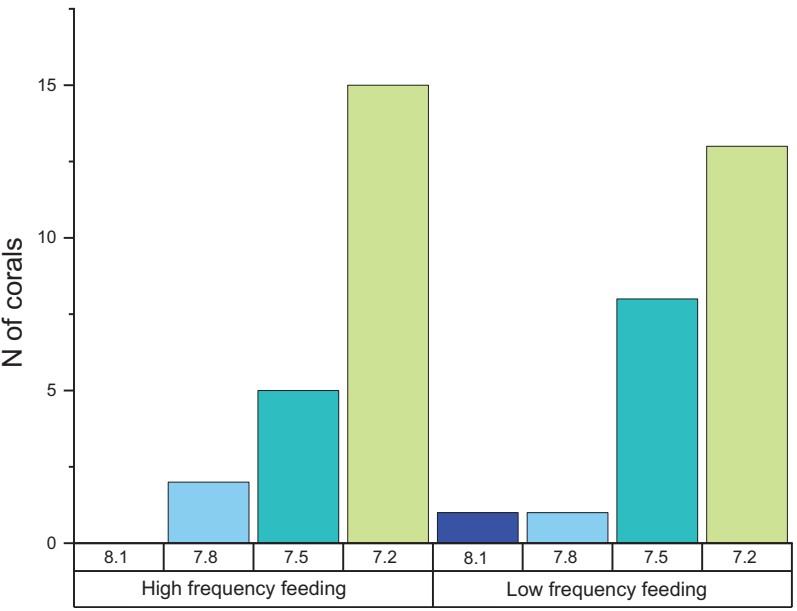

**Figure 5 Histogram of the number of corals that exhibited negative calcification rates in each treatment.** The total number of polyps was 152, and the total number of those that exhibited negative calcification was 48.

Our evidence for a negative impact of OA on coral calcification agrees with recent work investigating *L. pertusa* (*Gómez et al., 2018*), but contrasts with earlier studies based on shorter incubations, which were mostly undertaken at relatively higher pH, rarely below ~7.7 and $\Omega_A$ ~0.9 (Table 6). For example, studies on *D. dianthus* and other CWCs such as *L. pertusa* and *M. oculata* (*Maier et al., 2009*, *2012*; *Form & Riebesell, 2012*; *Tittensor et al., 2010b*; *Carreiro-Silva et al., 2014*; *Hennige et al., 2014*; *Movilla et al., 2014*) revealed no effect of high $p$CO$_2$ levels on net calcification. This is in agreement with our observations of non-significant differences in skeletal growth of corals incubated at pH 8.1 and 7.8, but contrasts with our findings at pH 7.5 and below. This apparent resistance at low levels of acidification has been attributed to the relatively low energetic requirements for calcification (*Maier et al., 2016*) and a particularly high degree of internal pH up-regulation in the calcifying fluid in these species (*Anagnostou et al., 2012*; *McCulloch et al., 2012*). In fact, this robustness and pH tolerance of CWCs is already evident from their natural habitat distribution. For example, *D. dianthus* corals, as a cosmopolitan species, have been found living over a wide range of pH levels in the Patagonian Fjords (8.1 to 7.4, *Jantzen et al., 2013b*; *Fillinger & Richter, 2013*) as well as in waters below the aragonite saturation horizon (*Anagnostou et al., 2011*; *Thresher et al., 2011*; *Jantzen et al., 2013b*). However, as mentioned above, pH and $\Omega_A$ levels could potentially reach rather extreme low levels in the near future, in the range of those experimented with in our study, particularly in the deep waters where CWCs develop. In the deep Pacific Ocean, for example, projections using the most pessimistic emission scenario RCP8.5 reach values of $\Omega_A$ below 0.5 by the end of the century (*Ciais et al., 2013*), and in the North Atlantic Ocean, the aragonite saturation horizon is expected to shoal

**Table 6 Summary of recent studies of Ocean Acidification effects on cold-water coral species from the world oceans.**

| Species | Area | Depth (m) | Incubation time | pH (total scale) | TA (µmol kg$^{-1}$) | $\Omega_A$ | $pCO_2$ (ppm) | Calcification rate | Reference |
|---|---|---|---|---|---|---|---|---|---|
| D. dianthus | Chilean Patagonia | 18–23 | 14 months | 8.12 ± 0.01 | 2,537 ± 4 | 2.79 ± 0.04 | 360 ± 10 | 0.309 ± 0.39 (mg CaCO₃ g$^{-1}$d$^{-1}$) | This study |
| D. dianthus | Chilean Patagonia | 18–23 | 14 months | 7.73 ± 0.05 | 2,541 ± 4 | 1.31 ± 0.13 | 1,030 ± 140 | 0.163 ± 0.31 (mg CaCO₃ g$^{-1}$d$^{-1}$) | This study |
| D. dianthus | Chilean Patagonia | 18–23 | 14 months | 7.50 ± 0.02 | 2,539 ± 4 | 0.79 ± 0.04 | 1,730 ± 100 | 0.305 ± 0.30 (mg CaCO₃ g$^{-1}$d$^{-1}$) | This study |
| D. dianthus | Chilean Patagonia | 18–23 | 14 months | 7.25 ± 0.02 | 2,540 ± 4 | 0.45 ± 0.02 | 3,180 ± 190 | −0.170 ± 0.31 (mg CaCO₃ g$^{-1}$d$^{-1}$) | This study |
| D. dianthus | Chilean Patagonia | 18–23 | 14 months | 8.10 ± 0.03 | 2,537 ± 4 | 2.67 ± 0.13 | 384 ± 34 | 0.055±0.99 (mg CaCO₃ g$^{-1}$d$^{-1}$) | This study |
| D. dianthus | Chilean Patagonia | 18–23 | 14 months | 7.73 ± 0.05 | 2,541 ± 4 | 1.31 ± 0.12 | 1,010 ± 130 | 0.105±0.15 (mg CaCO₃ g$^{-1}$d$^{-1}$) | This study |
| D. dianthus | Chilean Patagonia | 18–23 | 14 months | 7.52 ± 0.01 | 2,539 ± 4 | 0.82 ± 0.02 | 1,616 ± 37 | 0.017±0.13 (mg CaCO₃ g$^{-1}$d$^{-1}$) | This study |
| D. dianthus | Chilean Patagonia | 18–23 | 14 months | 7.24 ± 0.02 | 2,541 ± 4 | 0.44 ± 0.02 | 3,190 ± 190 | −0.320 ± 0.31 (mg CaCO₃ g$^{-1}$d$^{-1}$) | This study |
| D. dianthus | Adriatic sea | 430 | 8 months | 7.96 | | | 380 | 1.3 (µmol CaCO3 cm$^{-2}$ d$^{-1}$) | Gori et al. (2016) |
| D. dianthus | Adriatic sea | 430 | 8 months | 7.92 | | | 750 | 1.8 (µmol CaCO3 cm$^{-2}$ d$^{-1}$) | Gori et al. (2016) |
| D. dianthus | Adriatic sea | 430 | 8 months | 7.97 | | | 380 | 0.5 (µmol CaCO3 cm$^{-2}$ d$^{-1}$) | Gori et al. (2016) |
| D. dianthus | Adriatic sea | 430 | 8 months | 7.9 | | | 750 | 0.7 (µmol CaCO3 cm$^{-2}$ d$^{-1}$) | Gori et al. (2016) |
| D. dianthus | NW Mediterranean | 250 | 11 months | 7.81 | 2,543 ± 11 | 1.6 | 810 ± 53 | 0.6 (mg CaCO3 g$^{-1}$d$^{-1}$) | Movilla et al. (2014) |
| D. dianthus | E Mediterranean | 300 | 11 months | 7.81 | 2,536 ± 14 | 1.6 | 810 ± 53 | 1.06 (mg CaCO3 g$^{-1}$d$^{-1}$) | Movilla et al. (2014) |
| D. dianthus | Chilean Patagonia | 20 | 2 weeks | | | | | 0.25 ± 0.18 (% d$^{-1}$) | Jantzen et al. (2013a) |
| D. dianthus | Chilean Patagonia | 20 | 2 weeks | | | | | 0.09 ± 0.08 (% d$^{-1}$) | Jantzen et al. (2013a) |
| D. dianthus | Azores, NE Atlantic | 450 | 8 months | 7.7 | 2,389.8 ± 20.4 | 1.16 | 997 | 0.003 ± 0.004 (% d$^{-1}$) | Carreiro-Silva et al. (2014) |
| D. dianthus | Mediterranean | 690 | 5 days | | | 1.8 ± 0.2 | 563 ± 57 | 0.011 ± 0.007 (% d$^{-1}$) | Maier et al. (2012) |
| L. pertusa | California Margin | 300 | 7 weeks | 7.89 ± 0.04 | 2,282 ± 30 | 1.5 | 580 ± 8 | 0.02 ± 0.009 (% d$^{-1}$) | Gómez et al. (2018) |
| L. pertusa | California Margin | 300 | 7 weeks | 7.62 ± 0.06 | 2,305 ± 66 | 0.8 | 1,159 ± 34 | −0.010 ± 0.014 (% d$^{-1}$) | Gómez et al. (2018) |

| Species | Area | Depth (m) | Incubation time | pH (total scale) | TA (μmol kg⁻¹) | $\Omega_A$ | $pCO_2$ (ppm) | Calcification rate | Reference |
|---|---|---|---|---|---|---|---|---|---|
| *L. pertusa* | Norway | 145–220 | 6 months | 7.948 ± 0.47 | 1,962.8 ± 170.0 | 1.37 ± 0.19 | 430 ± 101 | 0.006 ± 0.004 (% d⁻¹) | *Büscher, Form & Riebesell (2017)* |
| *L. pertusa* | Norway | 145–220 | 6 months | 7.710 ± 0.056 | 2,221.5 ± 0.1 | 0.93 ± 0.14 | 899 ± 101 | 0.001 ± 0.003 (% d⁻¹) | *Büscher, Form & Riebesell (2017)* |
| *L. pertusa* | Norway | 145–220 | 6 months | 7.916 ± 0.031 | 1,885.10 ± 112.8 | 1.44 ± 0.14 | 448 ± 37 | 0.010 ± 0.004 (% d⁻¹) | *Büscher, Form & Riebesell (2017)* |
| *L. pertusa* | Norway | 145–220 | 6 months | 7.903 ± 0.052 | 1,603.78 ± 384.6 | 1.16 ± 0.40 | 392 ± 66 | 0.017 ± 0.006 (% d⁻¹) | *Büscher, Form & Riebesell (2017)* |
| *L. pertusa* | Norway | 145–220 | 6 months | 7.692 ± 0.076 | 1,982.8 ± 62.2 | 0.93 ± 0.19 | 859 ± 133 | 0.007 ± 0.006 (% d⁻¹) | *Büscher, Form & Riebesell (2017)* |
| *L. pertusa* | Norway | 145–220 | 6 months | 7.698 ± 0.057 | 1,888.20 ± 207.3 | 0.88 ± 0.10 | 804 ± 172 | 0.009 ± 0.003 (% d⁻¹) | *Büscher, Form & Riebesell (2017)* |
| *L. pertusa* | Norway | 70–155 | 2 weeks | 7.9 | 2,312 ± 21 | 1.38 ± 0.08 | 579 ± 41 | 0.067 ± 0.016 (% day⁻¹) | *Georgian et al. (2016)* |
| *L. pertusa* | Norway | 70–155 | 2 weeks | 7.75 | 2,312 ± 29 | 1.00 ± 0.05 | 845 ± 61 | 0.056 ± 0.015 (% day⁻¹) | *Georgian et al. (2016)* |
| *L. pertusa* | Norway | 70–155 | 2 weeks | 7.6 | 2,309 ± 26 | 0.74 ± 0.07 | 1,208 ± 132 | 0.048 ± 0.011 (% day⁻¹) | *Georgian et al. (2016)* |
| *L. pertusa* | Gulf of Mexico | 450–500 | 2 weeks | 7.9 | 2,333 ± 16 | 1.51 ± 0.10 | 552 ± 42 | 0.036 ± 0.007 (% day⁻¹) | *Georgian et al. (2016)* |
| *L. pertusa* | Gulf of Mexico | 450–500 | 2 weeks | 7.75 | 2,334 ± 17 | 1.07 ± 0.07 | 831 ± 54 | 0.0176 ± 0.012 (% day⁻¹) | *Georgian et al. (2016)* |
| *L. pertusa* | Gulf of Mexico | 450–500 | 2 weeks | 7.6 | 2,331 ± 15 | 0.80 ± 0.05 | 1,165 ± 76 | 0.009 ± 0.003 (% day⁻¹) | *Georgian et al. (2016)* |
| *L. pertusa* | Gulf of Mexico | 385–400 | 10 months | 7.90 ± 0.06 | 2,320 ± 85 | 1.47 ± 0.17 | | 0.011 ± 0.004 (% day⁻¹) | *Lunden et al. (2014)* |
| *L. pertusa* | Gulf of Mexico | 385–400 | 10 months | 7.80 ± 0.07 | 2,352 ± 32 | 1.18 ± 0.18 | | 0.01 ± 0.003 (% day⁻¹) | *Lunden et al. (2014)* |
| *L. pertusa* | Gulf of Mexico | 385–400 | 10 months | 7.67 ± 0.16 | 2,371 ± 9 | 0.97 ± 0.40 | | −0.007 ± 0.002 (% day⁻¹) | *Lunden et al. (2014)* |
| *L. pertusa* | Gulf of Mexico | 390–550 | 10 months | 7.90 ± 0.08 | 2,316±23 | 1.47 ± 0.23 | | 0.039 ± 0.01 (% day⁻¹) | *Lunden et al. (2014)* |
| *L. pertusa* | Gulf of Mexico | 390–550 | 10 months | 7.78 ± 0.04 | 2,340 ± 66 | 1.11 ± 0.1 | | −0.01 ± 0.004 (% day⁻¹) | *Lunden et al. (2014)* |
| *L. pertusa* | Gulf of Mexico | 390–550 | 10 months | 7.67 ± 0.1 | 2,309 ± 24 | 0.92 ± 0.23 | | −0.008 ± 0.003 (% day⁻¹) | *Lunden et al. (2014)* |
| *L. pertusa* | Mediterranean | 500 | 5 days | | | 2.5 ± 0.1 | 389 ± 10 | 0.033 ± 0.011 (% d⁻¹) | *Maier et al. (2012)* |
| *L. pertusa* | Mediterranean | 500 | 5 days | | | 2.6 ± 0 | 378 ± 4 | 0.09 ± 0.020 (% d⁻¹) | *Maier et al. (2012)* |
| *L. pertusa* | Mediterranean | 267 | 5 days | | | 2.1 ± 0.4 | 441 ± 56 | 0.034 ± 0.013 (% d⁻¹) | *Maier et al. (2012)* |
| *L. pertusa* | Mediterranean | 500 | 5 days | | | 1.3 ± 0.3 | 614 ± 72 | 0.037 ± 0.036 (% d⁻¹) | *Maier et al. (2012)* |
| *L. pertusa* | Mediterranean | 500 | 5 days | | | 1.5 ± 0.8 | 863 ± 588 | 0.068 ± 0.068 (% d⁻¹) | *Maier et al. (2012)* |
| *L. pertusa* | NW Mediterranean | 260–500 | 9 months | 8.10 ± 0.02 | 2,483 ± 54 | 2.4 | 444 | 0.021 ± 0.017 (% d⁻¹) | *Maier et al. (2013)* |
| *L. pertusa* | NW Mediterranean | 260–500 | 9 months | 7.73 ± 0.02 | 2,477 ± 36 | 1.3 ± 0.08 | 969 ± 25 | 0.021 ± 0.037 (% d⁻¹) | *Maier et al. (2013)* |

(Continued)

| Species | Area | Depth (m) | Incubation time | pH (total scale) | TA (µmol kg⁻¹) | $\Omega_A$ | $p\mathrm{CO_2}$ (ppm) | Calcification rate | Reference |
|---|---|---|---|---|---|---|---|---|---|
| *L. pertusa* | North Sea | 109–150 | 24 h | 7.91 | 2,255 | 1.38 | 544 | 7.85 (µmol Kg⁻¹ h⁻¹) | *Maier et al. (2009)* |
| *L. pertusa* | North Sea | 109–150 | 24 h | 7.76 | 2,203 | 0.97 | 791 | 7.72 (µmol Kg⁻¹ h⁻¹) | *Maier et al. (2009)* |
| *L. pertusa* | Norway | 100–285 | 6 months | 8.02 | 2,392 ± 46 | 1.72 | 509 ± 32 | 0.0068 ± 0.00324 (% d⁻¹) | *Form & Riebesell (2012)* |
| *L. pertusa* | Norway | 100–286 | 6 months | 7.96 | 2,298 ± 73 | 1.36 | 604 ± 105 | 0.00178 ± (% d⁻¹) | *Form & Riebesell (2012)* |
| *L. pertusa* | Norway | 100–287 | 6 months | 7.82 | 2,232 ± 61 | 1.03 | 778 ± 112 | 0.00073 ± 0.0029 (% d⁻¹) | *Form & Riebesell (2012)* |
| *L. pertusa* | Norway | 100–288 | 6 months | 7.76 | 2,349 ± 79 | 0.93 | 982 ± 146 | 0.00127 ± (% d⁻¹) | *Form & Riebesell (2012)* |
| *L. pertusa* | North Atlantic | 141–167 | 21 days | 7.77 ± 0.01 | 2,141.6±1.2 | 1.03 | 750 | 0.0438 ± 1.6 (% d⁻¹) | *Hennige et al. (2014)* |
| *M. oculata* | Mediterranean | 434 | 10 weeks | | | 2.30 | 453 | 122.7–109.9 (µmol m⁻² h⁻¹) | *Maier et al. (2016)* |
| *M. oculata* | Mediterranean | 434 | 10 weeks | | | 1.00 | 805 | 121.7–110.9 (µmol m⁻² h⁻¹) | *Maier et al. (2016)* |
| *M. oculata* | Mediterranean | 434 | 10 weeks | | | 0.63 | 1,638 | −12.8 (µmol m⁻² h⁻¹) | *Maier et al. (2016)* |
| *M. oculata* | NW Mediterranean | 260–500 | 9 months | 8.10 ± 0.02 | 2,525 ± 7 | 2.9 ± 0.11 | 380 ± 12 | 0.017 ± 0.014 (% d⁻¹) | *Maier et al. (2013)* |
| *M. oculata* | NW Mediterranean | 260–500 | 9 months | 7.74 ± 0.02 | 250 ± 62 | 1.4 ± 0.06 | 947 ± 20 | 0.038 ± 0.057 (% d⁻¹) | *Maier et al. (2013)* |
| *B. elegans* | California | | 8 months | 8.02 ± 0.02 | | 2.1 ± 0.05 | 410 ± 21 | 10–50 mg dry weight | *Crook et al. (2013)* |
| *B. elegans* | California | | 8 months | 7.78 ± 0.03 | | 1.3 ± 0.1 | 770 ± 75 | 5–38 mg dry weight | *Crook et al. (2013)* |
| *B. elegans* | California | | 8 months | 7.59 ± 0.02 | | 0.9 ± 0.04 | 1,220 ± 80 | 5–40 mg dry weight | *Crook et al. (2013)* |

very quickly, rising on the order of 1,000–1,700 m in only three decades, exposing some of the largest and more diverse CWC reefs in the world to undersaturated seawater (*Perez et al., 2018*). It is at these more extreme levels of pH and $\Omega_A$ that we found clear negative effects of OA on *D. dianthus* growth.

## Effect of time on the expression of potential effects

In a previous experiment undertaken by our group with *D. dianthus* and *Dendrophyllia cornigera*, of shorter duration (11 months), we observed that *D. dianthus* polyps managed to grow during the first stages of the experiment but net calcification decreased strongly after 6 months of exposure to acidified waters (*Movilla et al., 2014*). This previous study, although performed at only two pH levels (8.1 and 7.8), revealed an effect of time in the appearance of potential effects, prompting the need to undertake long (multiple months) experiments.

In the present experiment, performed at four pH levels, incubation time was statistically significant but we did not observe similar effects of time exposure (Table 3). In our pH 7.8 treatment, which was the lowest pH in the experiment of *Movilla et al. (2014)*, and the pH at which they observed a lagged response, we did not observe a significant decrease in net calcification over the course of the entire experiment. This could be partly explained by the fact that seawater pH at the natural location of the corals in this study is close to 7.8 (*Jantzen et al., 2013b*). At the two low pH treatments however (pHs of 7.5 and 7.2), the negative effects in coral growth were observed rather quickly, during the first 4 months, as shown by the BW measurements (Table 3). At the most extreme pH of 7.2, retraction of the coenenchyme and dissolution of the base was also visually evident, already during the first months of the experiment.

The time lag observed by *Movilla et al. (2014)* until the effects of OA on calcification rates manifested, suggests that exposure time could be a key variable governing the calcification response of organisms to OA. In unfavorable low-pH conditions, CWCs may devote more energy to biomineralisation at the expense of other important metabolic functions. There are many studies pointing to physiological and metabolic responses of CWCs to aragonite undersaturation, including decreases in lipid content, skeletal density, respiration and food capture rates (*Bramanti et al., 2013*; *Hennige et al., 2014*; *Movilla et al., 2014*; *Georgian et al., 2016*; *Gori et al., 2016*), as well as changes in the expression of genes involved in calcification, cellular stress and immune defenses specifically in *D. dianthus* (*Carreiro-Silva et al., 2014*) and tropical corals (*Vidal-Dupiol et al., 2013*). All these different responses indicate that the negative effects of OA can occur at all organismal levels and appear at the very beginning of an exposure to OA. The effects on vital structural functions such as skeletal growth and its maintenance seem to become evident, however, at later stages, when other physiological processes have already been compromised.

As discussed by *Movilla et al. (2014)*, it is also possible that, due to specific life processes, different periods or seasons are more susceptible to OA stress than others. For example, during reproductive stages, energy balance may be focused on gamete production, maturation and spawning rather than calcification. Very recently, a first insight has been

obtained into the reproductive biology of *D. dianthus*, specifically for specimens of the Chilean Fjords, in the same region where the polyps for this experiment were collected (*Feehan, Waller & Häussermann, 2019*). In that study, a highly seasonal reproduction was reported, spawning at the end of austral winter and beginning gamete production in early austral spring. During our experiment, we did not find evidence of reproduction (although this was indeed observed after this experiment was completed, see below), so the energy expenditure related to this vital process may not have played a role, perhaps explaining the lack of time lag we observed in the growth responses.

## Modulating effect of feeding on OA stress

Despite a number of studies focusing on the factors explaining the distribution and biology of CWCs, often determined by the availability of food resources, knowledge of the feeding ecology of CWCs is still very limited. Feeding rate and food availability have been suggested to modulate the effects of acidification on coral growth in tropical species (*Comeau, Carpenter & Edmunds, 2013*; *Drenkard et al., 2018*). For CWCs, only a few studies to date have attempted to investigate the combined effects of changing seawater pH and food availability on coral growth, specifically in *M. oculata* and *L. pertusa*, which are the most well-studied CWC species (*Larsson, Lundälv & Van Oevelen, 2013*; *Maier et al., 2016*; *Büscher, Form & Riebesell, 2017*).

Our first observation of the combined effects of feeding frequency and pH was that, as we expected, higher feeding frequency had a positive effect on net calcification rates and the overall performance of corals at all pH treatments (Fig. 3). At pH 8.1 and 7.8, LF corals grew less than HF corals in general, but the balance between calcification and dissolution was positive. In the particular case of pH 7.5, this balance was clearly positive in the HF conditions, but remained close to zero in the LF treatments. In the HF treatment, the additional input of energy from feeding apparently allowed corals to maintain positive skeletal accretion in aragonite undersaturated conditions. This mitigating effect of HF on the response of CWCs to environmental stressors could partially explain the extraordinarily high densities of *D. dianthus* found in the highly productive waters of the Chilean Patagonia fjords, despite the low pH and aragonite undersaturation that characterizes these regions, particularly in deep areas (*Jantzen et al., 2013a*, *2013b*; *Fillinger & Richter, 2013*). At the more extreme values of pH 7.2, however, corals in our experiment were unable to compensate for the adverse effects of acidification regardless of the feeding regime. In both feeding regimes, the average growth was negative. Thus, even in regions with abundant food, the extreme cases of OA will have an important impact and may compromise these species.

Our results, revealing an apparent mitigation of OA effects in coral growth at pH 7.5 (but not at pH 7.2) contrast with two recent studies based on *M. oculata* colonies (*Maier et al., 2016*) and *L. pertusa* (*Büscher, Form & Riebesell, 2017*), where no mitigating effects of feeding were observed in calcification rates under acidified conditions. We speculate that, in the face of environmental pressures, differences exist in growth responses and survival strategies between three-dimensional reef builders such as *M. oculata*

and *L. pertusa* and solitary polyps like *D. dianthus*. Solitary polyps may have a larger degree of freedom in adapting their response, independent of their neighboring polyps.

Energy requirements for coral calcification have been discussed and constrained in a variety of physiological and mechanistic studies that have addressed the issue from multiple perspectives. Carbonic anhydrases (CA), for example, have been shown to play a crucial role in calcification, by catalyzing the interconversion of $CO_2$ and bicarbonate ions, regulating the internal pH and the supply of ions for calcification (*Zoccola et al., 2016* and references therein). Transcellular transport of $Ca^{2+}$ through the $Ca^{2+}$-ATPase channel is also key (*Cohen & McConnaughey, 2003*), as well as specific proteins at the site of initial calcification (*Mass et al., 2014*; *Drenkard et al., 2018*). These, together with the need to pump protons out of the calcifying space and through the boundary layer (*Jokiel, 2011*) are, among other processes, examples of energy demanding mechanisms that may add to the clear differentiation observed in coral growth between HF and LF treatments in the present experiment.

A second observation from our experiment is the high variability in calcification rates of corals under the HF treatment as compared to the LF treatment, although this was less evident at the lowest pH (Fig. 3). Similar patterns of increased growth variability in response to a more abundant or better quality nutrition can also be seen in other studies with tropical corals (*Osinga et al., 2011*; *Houlbréque et al., 2015*; *Conlan et al., 2017*) and other CWCs (*Büscher, Form & Riebesell, 2017*), although no emphasis or discussion on the reason for this was provided. According to *Höfer et al. (2018)*, *D. dianthus* is a generalist species, feeding on a variety of zooplankton prey, which displays higher ingestion rates the higher the prey abundance, with no apparent limit. Therefore, we expect corals in the same treatment to feed as much as possible. The variability observed in HF treatment could be explained by the different morphological features of the polyps (size, calyx diameter, tentacle crown size), that would allow some corals to feed more efficiently than others. In addition, a food-replete environment may allow corals to reorganize energy allocation in order to increase inter-polyp competition for resources and reproductive success as observed in *Crook et al. (2013)*. In this particular study, high food conditions also seemed to alleviate some of the stress associated with high $pCO_2$ in the CWC solitary species *Balanophyllia elegans*. Better nutrition not only favored calcification, but also increased the number of larvae released and reduced juvenile coral mortality by 5–15% (*Crook et al., 2013*). A possible role of energy expenditure for reproduction is supported by our observation that, after this experiment was completed, reproduction was observed in eight polyps, and that this occurred mostly in *D. dianthus* polyps reared under HF (7 out of 8).

Despite these variable responses, CWCs are known to be well adapted to the extremely varying nutritious conditions of their natural habitat, as attested by the significant correlation between zooplankton food density and capture rates (*Tsounis et al., 2010*; *Naumann et al., 2011*; *Larsson, Lundälv & Van Oevelen, 2013*; *Höfer et al., 2018*). This correlation points to a growing concern about projections for the future which, in general terms, indicate diminished primary productivity in many regions worldwide (*Bopp et al., 2013*; *Fu, Randerson & Keith Moore, 2016*; *Moore et al., 2018*) with

implications also for zooplankton dynamics, already negatively impacted by OA (*Cripps, Lindeque & Flynn, 2014*). In this situation, although the ecological and biogeochemical interactions among calcifying plankton, non-calcifying plankton and the resultant feedbacks on CWCs are complex and poorly understood (*Hays, Richardson & Robinson, 2005*), our work evidences the importance of food availability for the fate of *D. dianthus* communities under OA. For several specific areas, however, these pessimistic projections on primary production are not so clear or may even be reversed. In the Southern Ocean, for example, a region where CWCs are abundant (*Thresher et al., 2011*), there seems to be a much weaker increase in stratification, partly related to the poleward shift in westerly winds and associated increases in upwelling which may promote primary productivity (*Swart & Fyfe, 2012*; *Fu, Randerson & Keith Moore, 2016*). Therefore the future may not be as bleak for CWCs in these regions, even with a predicted shoaling of the aragonite saturation horizon (*Bostock et al., 2015*).

### Ontogenic responses to low pH

Recent studies measuring vertical profiles of pH in the coelenteron of tropical corals show that OA may challenge calcification rates in adults with a deeper coelenteron cavity because of increased difficulty transporting $H^+$ out of the coelenteron and into the surrounding seawater (i.e., regulating internal pH; *Yuan et al., 2018*). This contrasts with the results of the work presented here with CWCs, where smaller young polyps were more strongly affected as was also documented in many other acidification experiments (*Comeau et al., 2014*; *Holocomb et al., 2014*; *Movilla et al., 2014*, *2016*; *Maier et al., 2016*). Indeed, for those juvenile *D. dianthus* reared at pH 7.2, thriving may not have been possible if central portions of the polyp had been affected. Faster calcification, as described in benthic organisms, has generally been attributed to greater energy allocation focused on skeletal growth (*Cohen et al., 2009* and references therein) in order to overcome the strong selective pressure at early life stages. Faster calcification, however, also requires more rapid transport of protons ($H^+$) from the site of calcification (i.e., more energy) to maintain internal pH compared to slow growers (*Comeau, Cornwal & McCulloch, 2017*). Despite the importance of recruits and young adults to the maintenance of coral communities, few studies have investigated the effects of OA in juvenile stages. According to *Drenkard et al. (2018)*, juveniles of the zooxanthelate coral *Favia fragum* were unable to maintain calcification rates at pH 7.5–7.2. Since the allocation of energy is a key life history trait that sets the functional basis for maximizing the fitness of organisms, especially under resource limitation or physiological stress, more research is needed to investigate the effects of OA at key ontogenic stages crucial for the recruitment and sustainability of coral populations.

### CONCLUSIONS

The findings of this research provide insight into the sensitivity of CWCs, and especially the solitary species *D. dianthus*, to the combination of OA and changes in food supply. Our data from this long-term incubation experiment show the vulnerability of *D. dianthus* coral populations to low pH conditions over long periods of time, starting at pH 7.5

($\Omega_A = 0.80$), with a strong response at pH 7.2 ($\Omega_A = 0.45$) and young life stages. High frequency feeding had a positive impact on net calcification rates of polyps regardless of the seawater pH. The increased food availability resulted in high inter-polyp, intra-colony variability of skeletal growth rates, as observed in nature, whereas low frequency feeding homogenized coral growth in all polyps, which probably focused their energy use on the vital process of building and maintaining their skeleton. Although the fate of CWC populations in coming centuries is still uncertain, it will be a result of the synergies and antagonisms generated by multiple environmental factors. In this study we show that seawater pH and food availability will likely play a key role, at least for *D. dianthus*. Additional long-term multi-factorial experimental approaches are needed to investigate the vulnerability and fate of CWCs in the future, especially at critical developmental stages, where corals are suspected to be at highest risk, and which are essential for maintaining coral communities and their associated biodiversity under future $CO_2$ scenarios.

## ACKNOWLEDGEMENTS

We would like to thank the Huinay Station staff and especially to Ulrich Pörschmann for their help provided during our stay in Comau Fjord, including logistics, diving and laboratory assistance. We are also grateful to Juancho Movilla, Antonio Canepa and Rafel Coma for their recommendations and help with data treatment and statistics. We greatly appreciate the editor Erik Cordes and the two anonymous reviewers of this manuscript for their constructive comments.This is publication nr 170 of Huinay Scientific Field Station.

### Funding

This work was supported by projects GEODESMO (2014CL0020), funded by Consejo Superior de Investigaciones Científicas (CSIC), Fundación Endesa y Fundación San Ignacio de Huinay and SCORE (CGL-2015-68194-R) funded by the Spanish Ministry of Science, Innovation and Universities, which included a Formación de Personal Investigador (FPI) PhD grant to Ariadna Martínez Dios. The funders had no role in study design, data collection and analysis, decision to publish, or preparation of the manuscript.

### Grant Disclosures

The following grant information was disclosed by the authors:
GEODESMO: 2014CL0020.
Consejo Superior de Investigaciones Científicas (CSIC), Fundación Endesa y Fundación San Ignacio de Huinay and SCORE: CGL-2015-68194-R.
Spanish Ministry of Science, Innovation and Universities, which included a Formación de Personal Investigador (FPI).

### Competing Interests

The authors declare that they have no competing interests.

## Author Contributions

- Ariadna Martínez-Dios analyzed the data, prepared figures and/or tables, authored or reviewed drafts of the paper, and approved the final draft.
- Carles Pelejero conceived and designed the experiments, authored or reviewed drafts of the paper, and approved the final draft.
- Àngel López-Sanz conceived and designed the experiments, performed the experiments, authored or reviewed drafts of the paper, and approved the final draft.
- Robert M. Sherrell conceived and designed the experiments, authored or reviewed drafts of the paper, and approved the final draft.
- Stanley Ko analyzed the data, authored or reviewed drafts of the paper, and approved the final draft.
- Verena Häussermann authored or reviewed drafts of the paper, helped with the logistics during the sampling of polyps, and approved the final draft.
- Günter Försterra authored or reviewed drafts of the paper, helped with the logistics during the sampling of polyps, and approved the final draft.
- Eva Calvo conceived and designed the experiments, authored or reviewed drafts of the paper, and approved the final draft.

## Field Study Permissions

The following information was supplied relating to field study approvals (i.e., approving body and any reference numbers):

The collection of animals for scientific purposes at the two sampling sites was approved by the sub-secretariat of fisheries and farming within the Chilean Ministry of Economy, Development & Tourism (ref. 1760).

CITES permit 14CL000006WS.

## Data Availability

The data on weight, calcification rate and mortality of *D. dianthus* corals during the experiment is available as a Supplemental File.

## Supplemental Information

Supplemental information for this article can be found online at http://dx.doi.org/10.7717/peerj.8236#supplemental-information.

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
