# Peer review of "Effects of low pH and feeding on calcification rates of the cold-water coral Desmophyllum dianthus"

_PeerJ, doi:10.7717/peerj.8236_

## Round 0.1 · original submission · Minor Revisions

I think that all of the reviewers' comments are valid and should be addressed, although I do not feel that this constitutes a major revision. In particular, I would like to see more of the environmental conditions as well as the experimental conditions. Some attention is required to the statistical treatment of the data and their interpretation. Finally, the references suggested by reviewer #2 are relevant to the manuscript, as are a few other recent references, and I urge the authors to complete another search for these and to update their language on the amount of previous work in this area.

Reviewer 1 ·

Basic reporting

The manuscript is professionally structured, with sufficient literature references, figures, and tables, and contains the appropriate raw data.

Overall the manuscript is well formulated, however there were several sections in which the English used could be improved. Suggestions:

Line 30-32: Sentence is too long and unnecessarily wordy.
Line 39: There is an extra space after the words 'efforts have'
Line 51: Shoaling of the saturation state 'horizon'
Line 70: ROV should be ROVs (plural).
Line 76: Replace 'First studies' with 'Early studies'
Line 77, 92, 284, 390: Check to make sure scientific names are italicized (check through the whole manuscript).
Line 84: Consider rephrasing 'evidenced signs of stress', perhaps 'exhibited signs of stress'
Line 95: Replace 'and at two' with 'and under two'
Line 98: Replace 'especially at' with 'especially in'
Line 100: Replace 'at unusual shallow' with 'at unusually shallow'
Paragraph beginning line 170: Should be moved to immediately after (or combined with) the description of the pH regulation system (paragraph starting line 137).
Line 250: Figure 5 is referred to prior to Figure 4 in the main text.
Line 265: Replace 'being those' with 'those being'
Line 270: replace 'have undertaken' with 'performed'
Line 272: This sentence is too long, should be reworded for clarity.
Line 324: The part of this sentence beginning with 'we did not...' needs to be reworded.
Line 328-330: sentence has several grammatical errors, and may need to be reworded.
Line 337: remove 'or other stressors'
Line 341-343: sentence is unclear, needs rewording.
Paragraph starting on line 365: This seems like it would be better placed in the methods.
Line 372-375: Sentence is unclear - the current wording seems to run counter to the findings of the experiment.
Line 391: It is unclear what is meant by this sentence.

Experimental design

The experiment was well designed, and a logical expansion of previous work to explore original research. The research questions were well defined, and represent a meaningful contribution to the field.

In the methods section, the manuscript would be improved by the addition of the following information:

1) The authors report a general range of pH values for the Comau Fjord, however the range reported is quite large (7.4-8.2). They collect their specimens at two different sites, and do not report local measurements of pH or other potential differences between collection sites, nor if there was significant differences in response during the experiment of corals from one site vs. the other (at least not in the main text). Including this information would help place their study in a broader context.

2) The authors do not report the pH and feeding conditions that the specimens were subjected to during their 9 month acclimatization period, nor the initial pH of the experiment. These conditions may have had considerable effect on the differences in the onset of OA effects relative to other experiments (i.e. where the corals primed by high feeding conditions, or a lower initial pH?).

Validity of the findings

Overall, the manuscript is robust and shows clearly statistically sound results that represent a significant contribution to the field. There are however several improvements that need to be made to make the work robust:

1) Despite the assertion of the authors on line 256 that the correlation shown in figure 4 is both significant and strong, it does not appear as such. There is also no r value presented (which appears to be very low). Perhaps the outlying specimen at initial weight of ~22.5 is driving this relationship, or perhaps its an artifact of over-massaging the data. There is also no explanation given for assigning an exponential function to the data. I recommend removing this result from the manuscript, as I have concerns about the validity of conclusions drawn from it.

2) It is unclear what conclusion is being made in the sentence beginning on line 346. The argument references a follow up experiment (which is again referenced on line 419), which is unpublished. If the authors wish to use the results of this experiment to justify their conclusions, those results should be included in this paper, or published elsewhere and then used in support. As stated, the main findings of that experiment are unclear and therefore confuse the points being made - especially if using it to justify the difference in results from previous experiments.

Reviewer 2 ·

Basic reporting

I reviewed the paper “Effects of low pH and feeding on calcification rates of the CWC Desmophyllum dianthus”, which aimed to investigate the long-term response of this CWC to different levels of pH and aragonite saturation at different feeding regimes. Overall, Martinez-Dios et al., found that D. dianthus is vulnerable to low pH conditions and low food availability. This is a well performed study on the calcification and feeding of an important CWC, with clear and correct use of English language, informative and well-structured introduction that provides sufficient and updated background of the topic. However, some of the references are inaccurately cited and need closer look where appropriate. Specific comments will follow this review.

Some references are missing from the text as in line 61, and others do not correspond to what it is stated in the paragraph such as in lines 78-80. There is a mix-up between short-term and long-term incubation thorough the manuscript. I would suggest defining in the methods or intro section what is considered for the authors long-term because otherwise the discussion becomes confusing. Just to give a few examples, Gori et al (2016) and Carreiro-Silva et al (2014) are both long-term and should not be discussed as short-term experiments (at least from the perspective of experimental biology and CWC in particular) or Form and Riebesell (2012) is a long-term experiment and line 81, Gomez et al (2018) is a short-term experiment. The same applies for lines 314-316: The authors citing Movilla et al. (2011) state “This previous study, although performed only at two levels of pH (8.1 and 7.8), revealed an effect of time in the appearance of potential effects, prompting for the need to undertake long-term experiments”. Again, do the authors consider 11 months incubation short-term?

Lines 85-87 the authors stated that “Regarding other stressors in addition to OA, few studies have been carried out with CWCs, and only two have examined their response to low pH and food availability (Maier et al., 2016; Gómez et al., 2018). However, Georgian et al (2016), and Busher et al. (2017), are also studies performed with CWC and included feeding variables, which should be included in here.

The authors need to double check references that are missing from the reference section but are cited in the text, as in lines 332-333, line 453 or lines 357-361, just to mention a few.

Experimental design

Well-designed experiment with a state-of-the-art set-up methodology that goes well according to the questions and main goals of the study. This study is well replicated thus the inference is very robust. All the methods were clearly stated in order to be replicated by other researchers.

Considerations: Please provide more detail about the n, such as replication units (fragments replication vs tank replication). Based on the statistical results and DF I will assume you used single fragments for your statistical inference?

Please check for typos as in lines 133, 214, 390, 423, or better use of English as in lines 25 or 48. Check redaction as in lines 372-375, which needs some rewording.

Validity of the findings

Findings are robust with the novelty of being one of the longest incubations of low pH and feeding regimes performed for CWC corals, although to my concern there is an excessive use of this novelty thorough the document. It is true that this study is longer than any other performed on this species, nevertheless, it is not the “only” long-term study and that should be stated.

There is only one concern about the environmental data from the collection site, such as pH, omega aragonite and alkalinity. I believe the authors provided information in general about the range in some environmental variables in the Fjords based on other studies and taking into account the whole bathymetric range for the species. If possible, they should provide the environmental range of the collection site at the depth of the collection, since the discussion has an important ecological component and comparison between studies with different biogeographic settings. Are the environmental values from the Fjords comparable to the ones found in the Mediterranean? Could it have an influence in the response found in the present study with other D.dianthus from elsewhere?

I would suggest moving the table for the models (i.e. ANOVA) from supplementary and include it in results.

In general, all the results are sound, meaningful and conclusions are well defined and supportive to the findings of the study.

Additional comments

Lines 361-364:
I do not think it is necessary to bound around this argument that has been mentioned all over the document. I would suggest ending the paragraph in line 361.

Lines 365-367:
Authors state: “In order to simulate as well as possible the natural field conditions and to compare with
published literature, corals in this experiment were fed taking into account both the food type and the optimal prey size observed in natural habitats of D. dianthus”. I wonder if the authors characterized this variable in the collection sites.

Line 445: double check references in here. I do not think Georgian fits in the argument.

Table 4:
Please check accuracy of incubation time for studies shown in the table. Also, I noticed there are some studies published recently that are not included in the table (specially for L pertusa). Is it supposed to be an exhaustive search? If the authors want to include a table (which I consider useful), I would suggest focus only on the studies more relevant to this study, perhaps only those related to OA and feeding with D. dianthus

---

## Round 0.2 · accepted · Accept

You have addressed all of the reviewers' and my own concerns with the manuscript. The figures are much clearer and the reporting of tank conditions and statistics have been greatly improved. I am happy to approve this manuscript for publication and I look forward to seeing it published.